# Intratumoral Niches of B Cells and Follicular Helper T Cells, and the Absence of Regulatory T Cells, Associate with Longer Survival in Early-Stage Oral Tongue Cancer Patients

**DOI:** 10.3390/cancers14174298

**Published:** 2022-09-01

**Authors:** Chumut Phanthunane, Rebecca Wijers, Maria J. De Herdt, Senada Koljenović, Stefan Sleijfer, Robert J. Baatenburg de Jong, José Angelito U. Hardillo, Reno Debets, Hayri E. Balcioglu

**Affiliations:** 1Department of Medical Oncology, Erasmus MC Cancer Institute, 3015 GD Rotterdam, The Netherlands; chumut.pha@cra.ac.th (C.P.); r.wijers@erasmusmc.nl (R.W.); s.sleijfer@erasmusmc.nl (S.S.); j.debets@erasmusmc.nl (R.D.); 2Department of Otorhinolaryngology, Erasmus MC Cancer Institute, 3015 GD Rotterdam, The Netherlands; m.deherdt@erasmusmc.nl (M.J.D.H.); s.koljenovic@erasmusmc.nl (S.K.); r.j.baatenburgdejong@erasmusmc.nl (R.J.B.d.J.); j.hardillo@erasmusmc.nl (J.A.U.H.); 3Department of Medical Oncology, HRH Princess Chulabhorn College of Medical Science, Bangkok 10210, Thailand

**Keywords:** tumor microenvironment, multiplex in situ staining, B cell, oral cancer, follicular helper T cells, regulatory T cells, T helper-type 1 cells

## Abstract

**Simple Summary:**

The 5-year survival of patients with early-stage oral cancer remains at 80% despite advances in treatment. We have previously shown that the proximity between CD20 B cells and CD4 T cells in the invasive margin acts as an independent prognosticator in OSCC, represented by the so-called CD20 cluster score. However, its exact underlying cellular contexture is unknown. In this study, we demonstrated that the abundance of follicular helper T cells as well as the proximity between these cells and B cells were important hallmarks for patients with a high CD20 cluster score and long survival.

**Abstract:**

In early oral squamous cell carcinoma (OSCC), the occurrence of clusters between CD20 B cells and CD4 T cells in the invasive margin (IM) can be captured by using the CD20 cluster score, and is positively associated with patient survival. However, the exact contribution of different CD4 T cell subsets, as well as B cell subsets toward patient prognosis is largely unknown. To this end, we studied regulatory T cells ((Treg cells) FOXP3 and CD4), T helper-type 1 cells ((Th1 cells) Tbet and CD4), follicular helper T cells ((Tfh cells) Bcl6 and CD4), B cells (CD20), germinal center B cells ((GC B cells) BCL6 and CD20), and follicular dendritic cells ((fDCs) CD21) for their density, location, and interspacing using multiplex in situ immunofluorescence of 75 treatment-naïve, primary OSCC patients. We observed that Treg, Th1-, Tfh-, and GC B cells, but not fDCs, were abundantly present in the stroma as compared with the tumor, and in the IM as compared with in the center of the tumor. Patients with high CD20 cluster scores had a high density of all three CD4 T cell subsets and GC B cells in the stromal IM as compared with patients with low CD20 cluster scores. Notably, enriched abundance of Tfh cells (HR 0.20, *p* = 0.04), and diminished abundance of Treg cells (HR 0.10, *p* = 0.03), together with an overall short distance between Tfh and B cells (HR:0.08, *p* < 0.01), but not between Treg and B cells (HR 0.43, *p* = 0.28), were significantly associated with overall survival of patients with OSCC. Our study identified the prognostic value of clusters between CD20 B cells and Tfh cells in the stromal IM of OSCC patients, and enabled an improved understanding of the clinical value of a high CD20 cluster score, which requires validation in larger clinical cohorts.

## 1. Introduction

Oral squamous cell carcinoma (OSCC) is the most common subsite of head and neck cancer, and annually, it accounts for 650,000 new cases and 350,000 deaths [1]. The five-year overall survival rate for patients with early-stage OSCC has remained at approximately 80% despite the development of interdisciplinary multimodality treatments [2,3,4,5]. OSCC is generally a highly immune-infiltrated type of cancer, yet the extent of infiltration and cellular composition of the infiltrate are diverse, leading to significant heterogeneity regarding OSCC’s microenvironments [6,7]. Charting the spatial location of lymphocytes within tumor microenvironments (TMEs) enhances our understanding of this heterogeneity in relation to patient survival, and may identify contextual prognosticators, as has been successfully exemplified for multiple types of cancer, such as breast, colorectal, non-small cell lung (NSCLC), ovarian, urothelial, pancreatic ductal adenocarcinoma, and head and neck cancer [8,9,10,11].

In OSCC, T lymphocytes are the principal components of tumor-infiltrating lymphocytes (TILs), and high numbers of CD8 and CD4 T cells have both been reported to correlate with longer patient survival [8]. CD4 T cells are differentiated into multiple sub-lineages that may differently contribute to antitumor immune responses [12], and the exact impact of different CD4 T cell subsets is not yet clear in OSCC [13]. At least seven CD4 T cell subsets have been identified in TMEs, namely: T regulatory (Treg) cells, T follicular helper (Tfh) cells, T helper type 1 (Th1) cells, T helper type 2 cells, T helper type 9 cells, T helper type 17 cells, and T helper type 22 cells [14,15]. Among the identified CD4 T cell subsets, Treg, Tfh, and Th1 cells account for approximately half of the CD4 TILs in OSCC [6,16]. In addition to the T cells, B cells also account for up to one-third of all TILs. However, the prognostic role of these B cell subtypes is also not well established [17].

Using multiplex immunofluorescence staining in oral cancer, we have recently introduced the CD20 cluster score, which provides a measure of clusters of CD20 B cells in co-occurrence with CD4 T cells in the invasive margin (IM), and acts as a prognostic marker predicting overall survival (OS) and disease-free survival (DFS) in patients with early OSCC (HR 0.34 and 0.47, *p* = 0.001 and 0.019; respectively) [18]. These clusters of CD20 B cells might resemble tertiary lymphoid structures (TLSs) found in various cancers [19,20,21,22]. CD4 T cells, especially Treg and Tfh cells, are known orchestrators of TLSs under physiological, non-cancerous conditions [23,24]. In cancer tissues, however, the abundance of different CD4 T cell subsets in TLSs is largely unknown [25,26,27]. We hypothesized that a differential abundance of various CD4 T cell subsets in the IM of OSCC is critical to the clustering between CD20 B cells and CD4 T cells, and consequently critical to patients’ survivals.

In this study, using a cohort of 75 patients with early-stage oral tongue cancer and making use of multiplex in situ immunofluorescence, we assessed the quantities and spatial organizations of B cells and various CD4 T cell subsets, and determined their individual prognostic values.

## 2. Materials and Methods

### 2.1. Patient Cohort

This study included 75 patients with pathological T1-2 oral-tongue cancer without perineural invasion (PNI), who received standard-of-care treatments at the Erasmus Medical Center (EMC; Rotterdam, The Netherlands) between October 2007 and December 2015. All patients had histologically proven primary squamous cell carcinoma and underwent curative surgery without any perioperative treatment. In this study, we analyzed 75 patients, representing a subcohort of a previously reported total cohort of 138 patients [18], where sufficient material was available. Tumors were processed into formalin-fixed, paraffin-embedded (FFPE) tissue blocks for multiplex immune staining. Clinical history, pathological staging according to the UICC 8th edition, and at least a 5-year clinical follow-up were introduced into medical databases. The EMC Medical Ethics Committee approved the research protocol (MEC-2016-751) following “The Code of Conduct for Responsible Use” and “The Code of Conduct for Health Research” as stated by the Federation of Dutch Medical Scientific Societies.

### 2.2. Histopathological Analysis

The FFPE tissue blocks were sectioned onto glass slides and used for hematoxylin and eosin (H&E) staining. H&E stained sections were digitally scanned for high-resolution whole slide images (WSIs). Histological parameters, namely, grading of differentiation and lymphovascular invasion (LVI), were retrieved from medical databases and reviewed by pathologists using glass slides or WSIs.

### 2.3. Immunofluorescence Staining

Immunofluorescence (IF) in situ staining was performed with CD4, CD20, CD21, FOXP3, Tbet, BCL6, and CK antibodies and nuclear counterstaining with DAPI (AKOYA Biosciences, Marlborough, MA, USA). Staining was performed using 4 μm FFPE sections, as previously described by [18]. In brief, seven sequential rounds of staining were performed; each round included antigen retrieval with microwave treatment in buffer, blocking, primary antibody incubation, secondary antibody incubation, and subsequent incubation with tyramide signal amplification (TSA) plus fluorophore, and all with washing steps in between. Finally, sections were counterstained with spectral DAPI and mounted with Vectrashield fluorescent mounting medium (Vector Laboratories, Burlingame, CA, USA). The 7-color multiplex protocol was optimized and validated using archival OSCC tissue, and specifics are provided in Appendix A.

### 2.4. Digital Image Analysis

Following multiplex staining, whole sections were scanned (10× magnification) and, for each section, 16 higher resolution multispectral images (MSIs) (magnification 20×, area 690 × 516 µm, resolution 2 pixels/µm, pixel size 0.5 × 0.5 µm^2^) were obtained, namely, 8 MSIs in the IM and 8 MSIs in the center of the tumor with the use of Vectra 3.0 (Akoya Biosciences, Menlo Park, CA, USA), as previously described by [18]. The spectral unmixing of MSIs was performed using the inForm^®^ software (Akoya Biosciences, Menlo Park, CA, USA) to visualize markers of interest as well as autofluorescence. Subsequently, the MSIs were analyzed using the Tumor Microenvironment Analyzer (TME-A), a python-based application developed by H.E.B. (manuscript in preparation, see Appendix A). In short, the following 5 steps were employed: (1) foreground selection, (2) tissue segmentation (Appendix A), (3) nucleus detection and segmentation (Appendix A), (4) cell segmentation (Appendix A), and (5) phenotyping. The CD4 T cells were phenotyped either as the sum of all CD4+ cells or as individual subsets, such as FOXP3+ CD4+ (Treg cells), Tbet+ CD4+ (Th1 cells), BCL6+ CD4+ (Tfh cells), FOXP3+BCL6+ CD4+ (T follicular regulatory (Tfr) cells), and FOXP3-Tbet-BCL6- CD4+ (other CD4 cells). The CD20 B cells were phenotyped either as CD20+ B cells or as individual subsets, such as, BCL6+ CD20+ geminal center (GC) B cells. Follicular dendritic cells (fDC) were phenotyped as CD21+ cells.

### 2.5. Assessment of Cellular Densities and Networks

The densities of the phenotyped cells were calculated by dividing the number of cells with a certain phenotype by the total area of a given region, and were averaged per patient across all corresponding MSIs. Densities were evaluated in 4 distinct regions, namely, center tumor (C-T), center stroma (C-S), IM tumor (IM-T), and IM stroma (IM-S). Fractions of subsets of CD4 T cells and CD20 B cells were calculated by dividing the number of T cell and B cell subsets by the total number of T cells and B cells, respectively. Spatial relationships between cell types with certain phenotypes were studied using the center of cells and according to nearest neighbor analyses (NNAs). These NNA analyses included distances (in µm) from one cell type to the nearest other cell type, and the number of one cell type within 20 µm of the same or another cell type. All NNAs were averaged across all cells and averaged over the MSIs in the IM region per patient.

### 2.6. The CD20 Cluster Score

The CD20 cluster score captured the number of CD20 B cells within 20 µm of CD4 T cells (CD20 within CD4) as well as the number of CD20 B cells within 20 µm of other CD20 B cells (CD20 within CD20) in the IM-S region, as described previously by [18]. The two individual components of the CD20 cluster score were classified into high versus low using their respective median value as a cut-off, after which these two components were combined into a single ordinal variable that distinguished high (CD20 within CD4 high and CD20 within CD20 high) versus low scores (CD20 within CD4 low and CD20 within CD20 low, CD20 within CD4 high and CD20 within CD20 low, or CD20 within CD4 low and CD20 within CD20 high).

### 2.7. Quantification of Tertiary Lymphoid Structures

The tertiary lymphoid structures (TLSs) were manually quantified using both the whole slide H&E images and the MSIs. From the H&E images, TLSs were identified as aggregations of lymphocytes in the IM region (approximately >50 cells in a region of 0.3 mm^2^). From the MSIs, aggregations of lymphocytes were again quantified in the same manner, but now using cells phenotyped as CD20 B cells and CD4 T cells [18,28]. The TLS count was reported as the total number of TLSs per tumor.

### 2.8. Statistical Analysis

Overall survival (OS) was defined as the time from diagnosis to death from any cause. Cox proportional hazard regression models using the enter method were used to determine univariate hazard ratios toward OS. Variables with *p*-values < 0.1 were subsequently used for multivariate Cox modeling using the backward elimination likelihood ratio method. Categorical variables were presented by frequency and percentages, and continuous variables were presented by median values. Comparisons between categorical variables were performed by chi-square tests. Mann–Whitney U tests were used for comparing continuous variables and the CD20 cluster score. Wilcoxon signed-rank tests were used for comparing continuous variables in dependent populations (i.e., distances between B cells and T cell subsets, and number of B cells within 20 μm of T cell subsets). Correlations were evaluated using Spearman correlation. Differences were tested using two-sided tests, and *p*-values < 0.05 were considered to be statistically significant. Statistical analyses were performed using the SPSS 24.0 software (SPSS Inc., Chicago, IL, USA).

## 3. Results

### 3.1. The CD20 Cluster Score Predicts Longer OS in Early OSCC and Associates with High Abundance of CD4 T Cells, CD20 B Cells, and TLSs, but Not with CD21 Cells in the IM-S Region

We have previously reported on the CD20 cluster score, i.e., a metric that combines the number of CD20 B cells within 20 µm of CD20 B cells (CD20 within CD20) and the number of CD20 B cells within 20 µm of CD4 T cells (CD20 within CD4), which correlated with the overall survival of patients with early oral cancer [18] (Figure 1A,B). To understand how different subsets of immune cells contribute to the CD20 cluster score, and thus patient prognosis, we quantified densities and networks of regulatory T (Treg) cells, T helper-type 1 (Th1) cells, follicular helper T (Tfh) cells, B cells (CD20), germinal center B (GC B) cells, as well as follicular dendritic cells (fDCs) using a cohort of 75 patients with a median follow-up duration of 62 months. The CD20 cluster score, also tested in this cohort, significantly correlated with the 5-year survival (Table 1). In the univariate analysis, both the CD20 cluster score and patient age had prognostic value towards OS (Appendix A, HR 0.28, 95% CI 0.09–0.85, *p*-value 0.03 and HR 4.18, 95% CI 1.51–11.62, *p*-value 0.01, respectively). Similar to our previous report [18], the multivariate analysis demonstrated that a high CD20 cluster score predicted longer OS when adjusted for patient age and pathological tumor stage (Appendix A, HR 0.26, 95% CI 0.09–0.80, *p*-value 0.02). As a first step in our immune cell analysis, we assessed to what extent the CD20 cluster score was associated with abundances of CD4, CD20, or CD21-positive cells in the tumor center or the IM region, both subdivided into tumor and stroma regions (Appendix A). We observed that patients with high CD20 cluster scores were characterized by high densities of CD4 T cells in the IM-S and C-T regions as well as high densities of CD20 B cells in the IM-S and IM-T regions (Figure 2A,B). A comparison of all four regions showed that CD4 and CD20 lymphocytes were the most dense in the IM-S region (Figure 2A,B). There was no association observed between the CD20 cluster score and the density of CD21 cells (Figure 2C). Further, we assessed the presence of TLSs in these regions, being defined by the co-clustering of CD4 and CD20 cells (See Materials and Methods for details, and Appendix A for example images). Notably, we observed a significantly higher TLS count in patients with high CD20 cluster scores versus those with low CD20 cluster scores (Appendix A, median number of TLSs 4 vs. 2, *p*-value < 0.01). In contrast to the CD20 cluster score, no significant association was observed between TLSs and OS (Appendix A, HR 0.95, 95% CI 0.31–2.87, *p*-value 0.93).

### 3.2. Patients with High CD20 Cluster Scores Demonstrate High Abundance of Stromal Treg, Th1, or Tfh, as well as GC B Cells

To assess the abundance of defined CD4 T cell subsets, we used the expression of transcriptional factors, i.e., FOXP3, Tbet, and BCL6, as phenotypical markers for Treg, Th1, and Tfh cell subsets, respectively. To assess the abundance of defined B cell subsets, we used the expression of BCL6 (GC B) and CD21 on the B cells. Interestingly, among patients with high CD20 cluster scores, all the subsets of CD4 T cells and CD20 B cells were detected in the vicinity of the cluster of CD20 B cells (Figure 1B–D). Moreover, the patients with high CD20 cluster scores expressed higher densities of Tregs, Th1, and Tfh cells, as well as GC B and other B cells in the IM-S region (Figure 2D–G). Of note, the association between the CD20 cluster score and densities of Tfh or GC B cells were not restricted to the IM-S region (Figure 2D–F). In this study, we focused on the IM-S region, since the highest abundances of the above subsets (in fact, all CD4 T cell and CD20 B cell subsets) were observed in the IM-S region, and their abundance was related to the CD20 cluster score, which itself is defined in the IM region. Details regarding the associations between the CD20 cluster score and CD4 and CD20 subsets in all regions are presented in Appendix A.

Exploratory analysis of the FOXP3+BCL6+ CD4 cell subset (Appendix A), likely representing follicular regulatory helper T (Tfr) cells, revealed that the density of Tfr cells was significantly higher in patients with a high CD20 cluster score as compared with those patients with a low CD20 cluster score (Appendix A). However, this outcome was observed for all CD4 T cell subsets that express 2 or 3 transcription factors, i.e., in addition to Tfr, Tbet+BCL6+ CD4+, Tbet+FOXP3+ CD4+, and Tbet+BCL6+FOXP3+ CD4+, only fractions of Tbet+BCL6+ CD4+ and Tbet+BCL6+FOXP3+ CD4+ showed a significant difference between patients with high and low CD20 cluster scores (Appendix A).

### 3.3. Patients with a High CD20 Cluster Score Demonstrate Co-Clustering of GC B Cells and Tfh Cells

Next, we looked into cellular networks according to distances among CD4 T and CD20 B cell subsets. When performing nearest neighbor analyses (NNAs, see Materials and Methods for details) between CD20 B cells and each of the three CD4 T cell subsets, we demonstrated that in patients with high versus low CD20 cluster scores, these distances were shorter (Figure 3A). In addition, in patients with high versus low CD20 cluster scores, the numbers of CD20 B cells within 20 µm of each of the CD4 T cell subsets were higher (Figure 3B). The actual nearest neighbor distances between CD20 B cells and each of the CD4 T cell subsets were not significantly different (Appendix A). The numbers of CD20 B cells within 20 µm of the Treg and Th1 cells were higher than that of Tfh (Appendix A). Notably, when performing the NNAs using BCL6+ CD20 B cells (generally showing similar trends, Appendix A), the highest numbers of GC B cells were observed within 20 µm of the Tfh in patients with high CD20 cluster scores high (Appendix A). Given the scarcity of Tfr cells or any of the CD4 T cell subsets that express 2 or 3 transcription factors, it was not possible to quantify nearest neighbor interactions for a large portion of the cohort.

Collectively, the above analyses suggest that despite the relatively high number of all CD4 T cell subsets, it is the co-clustering between GC B cells and Tfh cells that drives the CD20 cluster score (depicted in Appendix A).

### 3.4. The Prognostic Value of the CD20 Cluster Score Is Impacted by Enriched Abundance of Tfh Relative to Treg and Short Distance between Tfh Cells and B Cells

Finally, we assessed the prognostic value of individual measures regarding abundance or network. Our data revealed that rather than absolute densities of different CD4 T cell subsets, their relative abundance provided significant association with the CD20 cluster score. In fact, patients with high CD20 cluster scores had longer OS in the case of a low fraction of Treg cells (Figure 4, HR 0.10, 95% CI 0.01–0.82, *p*-value 0.03) and a high fraction of Tfh cells (Figure 4, HR 0.20, 95% CI 0.04–0.94, *p*-value 0.04). Moreover, the fraction of Treg cells negatively correlated with that of Tfh cells but not Th1 cells (correlation coefficients −0.30 and −0.05, *p*-values 0.01 and 0.65, see Appendix A for an overview regarding Spearman’s correlations between fractions of CD4 T cell subsets). Regarding distances, we observed that patients with high CD20 cluster scores had longer OS in the case of a short distance between CD20 B and Tfh cells (Figure 4, HR 0.08, 95% CI 0.01–0.67, *p*-value 0.02), but not for a short distance between GC B and Tfh cells (HR 0.289, 95% CI 0.053–1.587, *p*-value 0.153). Survival differences according to the CD20 cluster score were not observed for subgroup analysis for Th1 cells (Figure 3), CD20 B cells and subsets, or CD21 cells (Appendix A). Regarding Tfr cells or any of the CD4 T cell subsets that express 2 or 3 transcription factors, we found no impact towards the prognostic value of the CD20 cluster score for either density or fraction of any of these phenotypes (Appendix A).

A summary of the above findings and an overview of the critical determinants of the prognostic value of the CD20 cluster score in early OSCC is given in Figure 5. In this figure, we outlined three scenarios regarding the CD20 cluster score according to the relative abundance of Tfh and Treg cells (Appendix A gives patient specific quantifications and Appendix A presents association with survival, with *p*-value of log rank for OS being 0.042). In the first scenario, the CD20 cluster score is low which, independent of the relative abundance of Tfh cells, yields a poor prognosis with a 5-year survival rate of 74% (43 patients from our cohort). In the second scenario, the CD20 cluster score is high and the relative abundance of Tfh cells is low, yielding an intermediate prognosis with a 5-year survival rate of 88% (16 patients). In the third and last scenario, the CD20 cluster score is high and the relative abundance of Tfh cells is high, yielding a favorable prognosis with a 5-year survival rate of 100% (16 patients).

## 4. Discussion

The number of intratumoral lymphocytes has been recognized as having a major impact on the clinical course of OSCC [9,13]. In fact, the prognostic value of B cells has been established in several solid cancers [19,20,22,29], with recent reports also highlighting their role as a prognosticator in OSCC [18,30]. In addition, the co-clustering of CD20 B cells and CD4 T cells, which is captured by the CD20 cluster score, has prognostic value in patients with early OSCC [18]. In this study, we assessed the abundances and cellular networks of 5 major CD4, CD20, and CD21 subsets to identify critical determinants of the prognostic value of the CD20 cluster score in early OSCC. Our findings show that the co-clustering of B cells and Tfh cells, and an enriched abundance of Tfh cells relative to Treg cells in the stromal compartment of the invasive margin of OSCC dictate the prognostic value of the CD20 cluster score.

The co-clustering of B and CD4 T cells (identified by the CD20 cluster score), which is associated with higher density of B and CD4 T cells in the IM-S region and longer survival, might negatively affect growth of cancer cells in several manners. Intratumoral B cells can play an important role in tumor control [31]. For instance, they can stimulate tumor-specific T cells through presentation of tumor-specific antigens [32], production of antitumor antibodies [29,33], and immunostimulatory cytokines [34,35]. Here, patients with high CD20 cluster scores generally presented a high abundance of GC B cells; however, we did not observe a significant correlation between the abundance of GC B cells and survival (data not shown). These GC B cells together with surrounding non-GC B cells have been reported to act against cancer cells [36]. In fact, recent work on breast cancer patients has demonstrated that the presence of intratumoral GC B cells was associated with longer survival [37,38]. Taken together, these data support that the spatial organization of B cells and its subsets is beneficial for an antitumor immune response in early OSCC.

When zooming in on CD4 T cell subsets, we found opposing roles for Tfh and Treg cells. We observed that the prognostic value of the CD20 cluster score was mostly impacted by the co-clustering of B cells and Tfh cells, as well as an enriched abundance of the latter cell type relative to Treg cells in the IM-S region. In lymphoid organs as well as cancers, Tfh cells are important regulators of antigen-specific B cell responses, since they produce the B cell chemoattractant CXCL13 to which B cells respond through its receptor CXCR5 [39]. It is interesting to note that, in lung cancer patients, CXCL13-high tumors were associated with more TLSs and correlated with longer OS as well as an improved response to PD-1 blockade [40,41]. Tfh cells also produce IL-21, a critical regulator for B cell proliferation and differentiation into antibody-secreting plasma and memory B cells [42]. In fact, the combined stimulation of the IL-21R and the B cell receptor (BCR) induces B cells to produce granzyme B, which further aids in the establishment of an antitumor immune response [43]. These actions exemplify the potential critical value of an enriched abundance of these cells towards the prognostic value of the CD20 cluster score. To what extent Treg cells, including Tfr cells, affect the recruitment and function of Tfh cells in patients is largely unknown [44,45]. It has been reported that Treg cells could suppress T cell responses by IL-2 deprivation [46]; however, they may promote Tfh responses since differentiation of the latter cells has been enhanced by diminished levels of IL2 [47]. Interestingly, in a mouse model of lung adenocarcinoma, Treg cells reduced the number and cellular density of TLSs, especially affecting numbers of CD4 and CD8 T cells [26]. In addition, in NSCLC patients, tumors with a high number of TLS-associated B cells and a low number of Treg cells were associated with a better prognosis [48]. Together, these data point to the negative role of Treg cells in the B cell-mediated antitumor response. Tfr cells, generally defined as CXCR5+FOXP3+BCL6±ICOS± CD4+ cells, have been detected intratumorally in various cancers with limited assessments of their prognostic value [49]. In the cohort studied here, differential Tfr density and fraction, quantified as FOXP3+BCL6+ CD4+ T cells, did not associate with the prognostic value of CD20 cluster score. This suggests that, rather than Tfr cells, Treg and Tfh cells are orchestrating the CD20—Tfh cell clustering.

Interactions between B and Tfh cells generally take place within the TLS, and the presence of TLSs in tumors has been related to both prognosis [50,51] and therapy response [20,52]. In this study, we demonstrated that, in contrast to a high CD20 cluster score, high numbers of TLSs did not correlate with OS. The CD20 cluster score did not associate with the abundance of CD21-positive cells and may represent lymphoid structures that harbor GC B cells, as well as non-GC B cells, without a typical TLS meshwork of fDCs. Such structures have been reported previously and termed: non-classical lymphoid structures [53], early TLSs (E-TLSs) [40], spatial lymphocyte organizations [54], or immature TLSs [55]. Our study results suggest that the CD20 cluster score provides an alternative to TLS assessment in early OSCC with improved prognostic value. Along this line, we showed that a four-marker staining (CD4, BCL-6, FOXP3, and CD20), together with NNA analyses, enabled the identification of a subset of patients with 100% 5-year survival versus those with 74% 5-year survival. The CD20 cluster score, when implemented in a diagnostic setting, would provide a clinical prognosticator for early OSCC.

Our study does show a number of limitations. First, the phenotyping of immune cells could have been performed according to a larger number of markers. For example, it is important to note that we selected the BCL6 marker to identify Tfh cells because it is a lineage-specific transcription factor with functions in the development of Tfh cells [56]. However, Tfh cells can also be identified by co-expression of PD-1, ICOS, and CXCR5 [56]. Moreover, in addition to Tfh cells and GC B cells, BCL6 can also be expressed by epithelial cancer cells [57,58], which may challenge the assessment of Tfh and GC B cells. Additionally, GC B cells solely defined by BCL6 positivity might not represent classical GC B cells, which are generally defined by CD20+, CD23+, activation-induced cytidine deaminase (AID)+, Ki-67+, and BCL6+ [42]. Second, studies that address mechanisms that underlie the formation of clusters between B and Tfh cells would further our understanding of the Tfh–B cell interaction [31]. To this end, we would advocate studies on CXCL13–CXCR5 as well as other co-inhibitory ligand-receptor pairs, such as PD-1–PD-L1, TIM-3–Galectin-9, LAG3–MHC class II, and CTLA-4–CD80/86. Indeed, CXCL13 has already been shown to be significantly upregulated in highly exhausted CD4 and CD8 T cells in melanoma, hepatocellular carcinoma, ovarian cancer, and NSCLC [41,59,60,61,62], and its receptor has been shown to be involved in immune cell aggregation and to be present on PD-1 blockade responsive exhausted CD8 T cells [63,64]. Initial interrogation of the TCGA database revealed that none of the abovementioned ligands and receptors have prognostic value for patients with pT1-T2, pN0-1, or M0 oral cancers (data not shown). A next and more conclusive step, also enabling the identification of the exact CD4 T cell subset that expresses such ligand-receptor pairs, would rely on the future use of spatial and single cell transcriptomics. In fact, such studies may yield markers or actionable targets to treat OSCC patients according to their CD20 cluster score. Such studies may yield markers or actionable targets to treat OSCC patients with low CD20 cluster scores. Lastly, the testing of the prognostic value of the CD20 cluster score in an independent cohort of treatment-naïve OSCC patients, as well as the testing of its predictive value in a cohort of OSCC patients treated with standard of care or immune checkpoint inhibitors, is needed to validate and extend our findings.

## 5. Conclusions

Our results demonstrate that proximity between Tfh and CD20 B cells, as well as enriched abundance of the former cell type relative to Treg cells in the invasive margin, critically determine the prognostic value of the CD20 cluster score for patients with early OSCC.

## Figures and Tables

**Figure 1 cancers-14-04298-f001:**
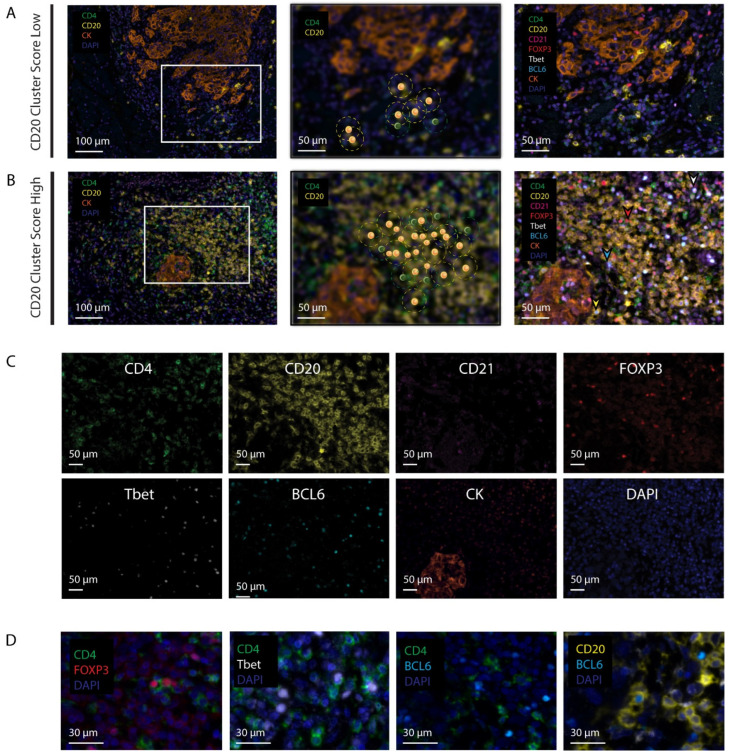
Multispectral images of CD20 cluster score low and high tumors. (**A**,**B**) Representative images of low CD20 cluster score (**A**) and high cluster score (**B**) tumors. (Left, magnification 20×, the multispectral images of CD4, CD20, CK, and DAPI; Middle, zoomed in, CD20 B cells within 20 µm radius from either CD4 T cells (CD20WCD4, green dotted line) or CD20 B cells (CD20WCD20, yellow dotted line) are highlighted; Right, accompanying multispectral images of CD4, CD20, CD21, FOXP3, Tbet, BCL6, CK, and DAPI); (**C**) monoplex images for individual markers of a high CD20 cluster score tumor; (**D**) zoomed-in images of double-positive lymphocytes (right, FOXP3+ CD4+ T cell (red arrowhead), Tbet+ CD4+ T cell (white arrowhead), BCL6+ CD4+ T cell (cyan arrowhead), and BCL6+ CD20+ B cell (yellow arrowhead)).

**Figure 2 cancers-14-04298-f002:**
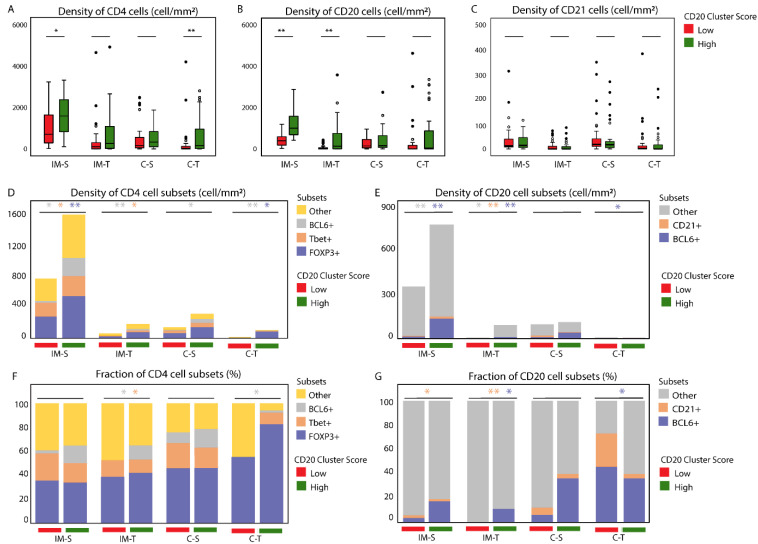
The CD20 cluster score associates with high densities of Treg, Th1 and Tfh, as well as GCB cells in the IM-S. (**A**–**C**) Box plots showing CD4 (A), CD20 (B), and CD21 (C) cell densities according to the CD20 cluster score in all 4 regions; (**D**,**E**) bar plots showing median densities of CD4 T cell subsets and CD20 B cell subsets according to the CD20 cluster score in all 4 regions; (**F**,**G**) the median fractions of CD4 and CD20 subsets according to the CD20 cluster score in all 4 regions. Statistical significance according to the Mann–Whitney U test is shown above individual plots, * *p* < 0.05, and ** *p* < 0.01. Colored asterisks indicate a difference between the high versus low CD20 cluster score for the corresponding cell subset. IM-S, invasive margin stroma, IM-T, invasive margin tumor, C-S, center stroma, and C-T, center tumor.

**Figure 3 cancers-14-04298-f003:**
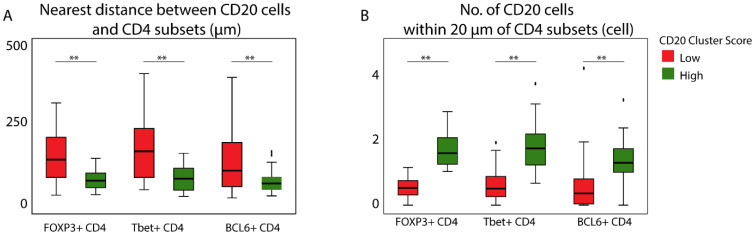
The CD20 cluster score associates with shorter distances between CD20 B cells and CD4 T cell subsets, and higher numbers of CD20 B cells within 20 µm of CD4 T cell subsets: (**A**,**B**) Boxplots showing the nearest distance between CD20 and CD4 subsets (**A**), and the number of CD20 B cells within 20 µm of CD4 T cell subsets (**B**), according to the CD20 cluster score in the IM-S region. Mann–Whitney U test was applied to test for significant differences, ** *p* < 0.01.

**Figure 4 cancers-14-04298-f004:**
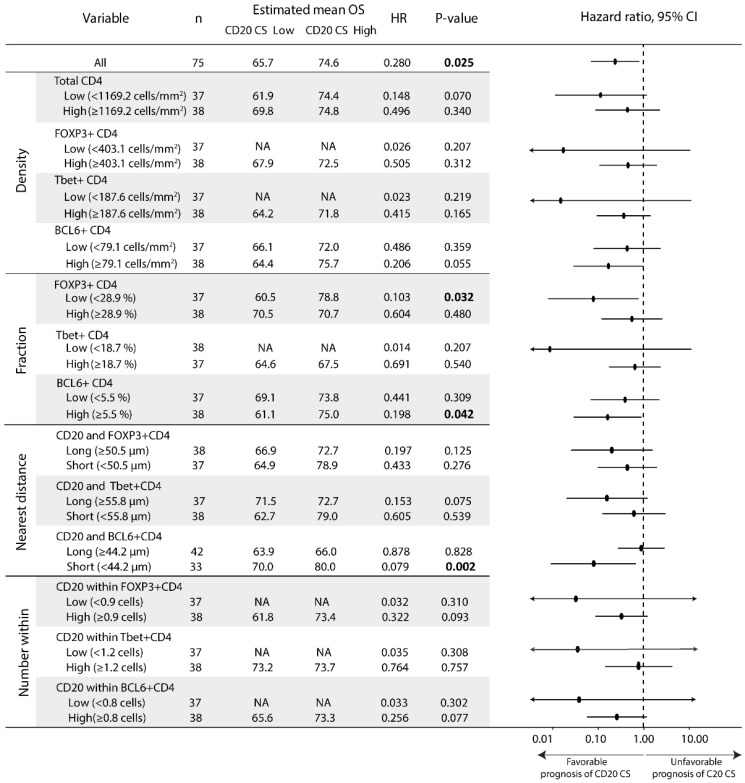
The prognostic value of the CD20 cluster score is impacted by a low fraction of Treg cells, a high fraction of Tfh cells, and a short distance between CD20 B and Tfh cells. Forest plot of subgroup analysis for the CD20 cluster score according to density and fraction of CD4 T cell subsets, as well as nearest distances to and numbers within 20 µm from B cells of CD4 T cell subsets. The estimated mean overall survival, HR, 95% CI, and *p*-value are shown for each variable; in the case of *p*-value < 0.05, this is highlighted in bold. Abbreviations: CD20 CS, CD20 cluster score; HR, hazard ratio; NA, non-applicable.

**Figure 5 cancers-14-04298-f005:**
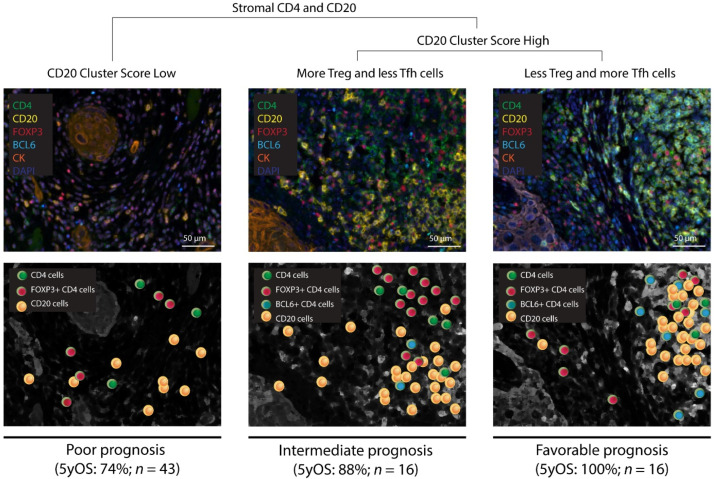
The prognostic value of the CD20 Cluster Score according to the relative abundance of Tfh and Treg cells. Three scenarios for patients with early OSCC showing different CD20 cluster scores and abundances of Tfh cells (BCL6+ and CD4+) relative to Treg cells (FOXP3+ and CD4+). Each scenario is accompanied by a representative multispectral image of a tumor with CD4 T cell or CD20 B cell subsets being highlighted. 5yOS, 5-year overall survival rate.

**Table 1 cancers-14-04298-t001:** Clinicopathological characteristics ^a^.

Variables	CD20 Cluster Score Low (*n* = 43)	CD20 Cluster Score High (*n* = 32)	*p*-Value
	Median (Interquartile range)		
Age (years)	63 (50–71)	62 (51–73)	0.945
	Number (%)		
Gender			0.481
Male	26 (60%)	16 (50%)	
Female	17 (40%)	16 (50%)	
Tumor (T) stage			0.420
pT1	19 (44%)	19 (59%)	
pT2	20 (47%)	10 (31%)	
pT3	4 (9%)	3 (10%)	
Nodal (N) stage			0.487
pN0	39 (91%)	30 (94%)	
pN1	4 (9%)	2 (6%)	
Stage			0.449
pStage 1	18 (42%)	18 (56%)	
pStage2	17 (40%)	9 (28%)	
pStage3	8 (18%)	5 (16%)	
Differentiation grade			0.159
Well differentiated	6 (14%)	10 (31%)	
Moderately differentiated	32 (74%)	18 (56%)	
Poorly differentiated	5 (12%)	4 (13%)	
Lymphovascular invasion			0.991
Absence	39 (91%)	29 (91%)	
Presence	4 (9%)	3 (9%)	
5-year survival			0.034
Alive	28 (65%)	28 (88%)	
Dead	15 (35%)	4 (12%)	
Locoregional recurrence			0.568
No recurrence	37 (86%)	28 (88%)	
Recurrence	6 (14%)	4 (12%)	

^a^ The table lists clinicopathological characteristics of the patients with high and low CD20 cluster scores. Statistical significance was tested between cohorts using a chi-square test, with the exception of age, where a Mann–Whitney U test was used. Staging was done according to the 8th edition of AJCC staging system. pT, pathological tumor stage; pN, pathological nodal stage; pStage, pathological stage.

## Data Availability

Raw imaging data, extended data tables, and analysis codes are available upon reasonable request. The remaining data are available within the article and Appendix A.

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
