# Peer review of "Intratumoral Niches of B Cells and Follicular Helper T Cells, and the Absence of Regulatory T Cells, Associate with Longer Survival in Early-Stage Oral Tongue Cancer Patients"

_cancers, 2022, doi:10.3390/cancers14174298_

Round 1
Reviewer 1 Report
Phanthunane et al present a succinct and well-presented analysis building upon previous work, indicating that the histological relationship between B cells and CD4+ T cells has prognostic value in oral squamous cell carcinoma. They report that the outcome of patients can be further predicted by additionally including and quantifying the presence of regulatory T cells and Tfh cells.
The paper is relatively easy to follow and the figures and conclusions are sound. I particularly appreciated Figure 5 to draw the paper together. I have made some suggestions below to enhance the quality of the paper.
Major comments:
- it is unclear to the reviewer whether this cohort represent a new cohort (i.e. a validatory sample) or a subset of the original study cohort from which the CD20 Cluster Score was derived. This should be clarified in the abstract and text.
- The assumption here is that the CD4 Tfh and CD20 B cells are cooperating to produce an effective anti-tumoural humoral response (alongside other pathways) - which may be disrupted in some patients by the effect/presence of Treg. As the authors briefly discuss in their conclusion, there are different lineages of regulatory T cells - most pertinantly to this study, regulatory T follicular helper cells (Tfr). With the combination of cell markers utilised, the authors should be able to assess for the presence of CD4+BCL6+FOXP3+ Tfr cells and these should be reported within their analyses.
Minor comments:
- The title feels 'clunky' and should be reworded for clarity.
- In the opening paragraph of the introduction, the authors refer to mortality rate with respect to OSCC outcomes. Survival rate is a generally preferred measure and I would suggest this is changed.
- The generic instructions for the introduction have been left in and should be removed.
- Is there a justification for the definition of Tertiary Lymphoid Structure used (>50 cells in 0.3mm^2)?
Author Response
Major comments:
1. It is unclear to the reviewer whether this cohort represents a new cohort (i.e. a validatory sample)
or a subset of the original study cohort from which the CD20 Cluster Score was derived. This
should be clarified in the abstract and text.
Response: We thank R#1 for pointing out this lack of clarity. The patient cohort used in this study,
consisting of 75 patients, is a subgroup of the total cohort of 138 patients used in Phanthunane C et
al., Oncoimmunol, 2021, where sufficient material was available. Along R1’s recommendation, we
have described the patient cohort in the revised Material and Methods.
2. The assumption here is that the CD4 Tfh and CD20 B cells are cooperating to produce an
effective anti-tumoural humoral response (alongside other pathways) - which may be disrupted in
some patients by the effect/presence of Treg. As the authors briefly discuss in their conclusion,
there are different lineages of regulatory T cells - most pertinantly to this study, regulatory T
follicular helper cells (Tfr). With the combination of cell markers utilised, the authors should be
able to assess the presence of CD4+BCL6+FOXP3+ Tfr cells and these should be reported within
their analyses.
Response: We concur with R#1 that Tfr may represent a subset of CD4 T cells that potentially
affects the prognostic value of the CD20 Cluster Score, and should be part of our analyses. To this
end, we have explored the density of FOXP3+BCL6+ CD4+ cells, as well as the fraction of these
cells of the total CD4 T cell population, and their impact towards the prognostic value of the CD20
Cluster Score. We observed that Tfr cells were mainly present in the IM-S region, present with a
median density of 21.8 cell/mm2, and represented 1.4% of all CD4 T cells. Interestingly, the density
of Tfr cells was significantly higher in patients with a high CD20 Cluster Score compared to those
patients with a low CD20 Cluster Score. However, this outcome was true for all CD4 T cell subsets
that express 2 or 3 transcription factors, i.e., besides Tfr, the phenotypes: Tbet+BCL6+ CD4+;
Tbet+FOXP3+ CD4+; and Tbet+BCL6+FOXP3+ CD4+. Moreover, we found no impact towards
the prognostic value of the CD20 Cluster Score for neither density nor fraction of any of these
phenotypes. Given the scarcity of Tfr cells or any of the CD4 T cell subsets that express 2 or 3
transcription factors, it was not possible to quantify nearest neighbor interactions for a large
portion of the cohort. The above results are displayed in Supplementary Figure 5 of the revised
manuscript. Additionally, we have updated text in the Materials and Methods, Results and
Discussion sections of the revised manuscript.
Minor comments:
- The title feels 'clunky' and should be reworded for clarity.
4
Response: The title is reworded in the revised manuscript as follows: “Intra-tumoral niches of B
cells and follicular helper T cells, and absence of regulatory T cells, associate with longer survival
in early-stage oral-tongue cancer patients”.
- In the opening paragraph of the introduction, the authors refer to mortality rate with respect to
OSCC outcomes. The survival rate is a generally preferred measure and I would suggest this is
changed.
Response: We have adapted the opening paragraph of the Introduction of the revised manuscript
accordingly.
- The generic instructions for the introduction have been left in and should be removed.
Response: The generic instructions for Introduction have been removed in the revised manuscript.
- Is there a justification for the definition of Tertiary Lymphoid Structure used (>50 cells in
0.3mm^2)?
Response: There is no standardized method available for TLS quantification. Our methodology
used to quantify TLS approximates the presence of PNAd+ HEV as we have previously reported for
10 patients [doi:10.1080/2162402X.2021.1882743], where we observed that a minimum total of 50
lymphocytes, particularly comprising CD20 B cells and CD4 T cells, well associated with lymphoid
structures according to PNAd+ HEV. In addition, our methodology is also in agreement with TLS
assessment in lung cancer, where in identified TLS a minimal number of 45 lymphocytes (median
453, mean 620.8, max 2936), and a minimum area of 0.06mm2 (median 0.0358 mm2, mean 0.0483
mm2, max 2.3mm2) were observed [doi:10.1371/journal.pone.0256907].

Reviewer 2 Report
In their manuscript, Phanthunane el. al. present proximity of T follicular helper cells and B cells at the invasive margin of OSCC tumors and shows the correlation of high CD20 cluster score with better overall survival in the patients. Overall, the manuscript is well-written and addresses B cells’ interesting and potentially significant role in anti-tumor immune responses.
Major comments:
1. It has been shown that the exhausted or dysfunctional CD8+ and CD4+ TILs frequently express CXCL13, which suggests the need for B cells (PMID: 29892065, 30872264, 32359441 & 35393541). Such interactions can culminate in the formation of TLSs. Although the authors have discussed the study’s limitations, it will be helpful to show the expression levels of inhibitory or exhaustion markers on the TfH cells. Optional: In addition, the expression of CXCL13 by TfH cells will help to understand their relationship with CD20+ B cells.
2. The authors discussed different types of CD4+ T except for exhausted CD4+ T cells in the tumor microenvironment. Although the discussion is well written it is recommended that there be an additional mention about T cell exhaustion incorporated into it.
Author Response
Major comments:
1. It has been shown that the exhausted or dysfunctional CD8+ and CD4+ TILs frequently express CXCL13, which suggests the need for B cells (PMID: 29892065, 30872264, 32359441 & 35393541). Such interactions can culminate in the formation of TLSs. Although the authors have discussed the study’s limitations, it will be helpful to show the expression levels of inhibitory or exhaustion markers on the TfH cells. Optional: In addition, the expression of CXCL13 by TfH cells will help to understand their relationship with CD20+ B cells.
Response: We thank R#2 for highlighting the potential relevance of CXCL13+CD4+ T cells, particularly exhausted CXCL13+ Tfh cells, in relation to CD20+ B cells and the prognostic value of the CD20 Cluster Score. We fully align with R#2 that further understanding and clinical exploitation of the Tfh – B cell interaction would greatly benefit from studies into ligand-receptor pairs, such as CXCL13 – CXCR5, PD-1 – PD-L1, TIM-3 – Galectin-9 etc. Indeed, CXCL13 has
already been shown to be significantly upregulated in highly exhausted CD4 and CD8 T cells in melanoma, hepatocellular carcinoma, ovarian cancer and non-small cell lung cancer [refs PMID: 21555851, 30595452, 28622514, 30872264, 29892065], and its receptor has been shown to be involved in immune cell aggregation and to be present on PD-1 blockade responsive exhausted CD8 T cells [refs PMID: 27501248, 27501245]. To provide a first step in such understanding, we have interrogated the prognostic value of the ligand CXCL13, its receptor CXCR5, and various pairs of co-inhibitory ligands and receptors, such as PD-L1, PD-1, CD80, CD86, CTLA-4, Galectin-9, TIM-3, MHCII, and LAG3 using the TCGA database for patients with pT1-T2, pN0-1, M0 oral cancers (n=103). None of the listed genes provided significant prognostic value. Given the small size of the patient cohort, subgroup analyses of these genes in relation to B cell or CD4 T cell subset-enriched or depleted tumors were not feasible. A second and more conclusive step in such understanding would rely on the use of spatial and single cell transcriptomics. Authors are of the opinion that such additional studies, despite their relevance, would be beyond the scope of the
manuscript presented here. Nevertheless, and along R#2’s recommendation, we have revised the Discussion section, and explicitly pointed to a possible contribution of CXCL13+ Tfh to the prognostic value of the CD20 Cluster Score and discussed how future studies into ligand-receptor pairs could enhance our understanding and clinical exploitation of the Tfh – B cell interaction.
2. The authors discussed different types of CD4+ T except for exhausted CD4+ T cells in the tumor microenvironment. Although the discussion is well written it is recommended that there be an additional mention about T cell exhaustion incorporated into it.
Response: We thank R#2 for this comment. We have addressed exhaustion of CD4 T cells and their
relation to the CD20 Cluster Score as part of our response to the above concern.

Round 2
Reviewer 1 Report
I have no further comments. My congratulations to the authors.
Reviewer 2 Report
The authors have satisfactorily addressed most of my concerns